# Yeast One-Hybrid Screening for Transcription Factors of IbbHLH2 in Purple-Fleshed Sweet Potato

**DOI:** 10.3390/genes14051042

**Published:** 2023-05-05

**Authors:** Danwen Fu, Yahui Chen, Feng Gao

**Affiliations:** 1Guangdong Provincial Key Lab of Biotechnology for Plant Development, School of Life Sciences, South China Normal University, Guangzhou 510631, China; 2Institute of Nanfan & Seed Industry, Guangdong Academy of Sciences, Guangzhou 510310, China

**Keywords:** anthocyanin biosynthesis, IbERF1, IbERF10, transcription regulators

## Abstract

The transcription factor IbbHLH2 has been identified as involved in the biosynthesis of anthocyanins in purple-flesh sweet potatoes. However, little is known about the upstream transcription regulators of the promoter of *IbbHLH2* in terms of their involvement in anthocyanin biosynthesis. For this study, the transcription regulators of the promoter of *IbbHLH2* were screened via yeast one-hybrid assays in purple-fleshed sweet potato storage roots. Seven proteins, namely IbERF1, IbERF10, IbEBF2, IbPDC, IbPGP19, IbUR5GT, and IbDRM, were screened as upstream binding proteins of the promoter of *IbbHLH2*. The interactions between the promoter and these upstream binding proteins were verified using dual-luciferase reporter and yeast two-hybrid assays. Furthermore, the gene expression levels of transcription regulators, transcription factors, and structural genes involved in the anthocyanin biosynthesis of different root stages of purple and white-fleshed sweet potatoes were analyzed via real-time PCR. The obtained results indicate that IbERF1 and IbERF10 are key transcription regulators of the promoter of *IbbHLH2* and are involved in anthocyanin biosynthesis in purple-fleshed sweet potatoes.

## 1. Introduction 

Anthocyanins are water-soluble flavonoid compounds that are widely present in the vacuoles of most plants [1]. Anthocyanins are a kind of plant pigment commonly found in the roots and tubers, petals, leaves, and fruit of plants, giving them red, pink, blue, and purple colors, which can be assessed as a sign of fruit ripening [2]. Studies have shown that anthocyanins have important biological significance for the adaptation of plants to their environment, as these compounds can improve their photoprotection ability, cold and drought resistance, and antibacterial and antioxidant resistance to biotic and abiotic stresses [3]. Anthocyanins are heavily involved in the protective mechanisms corresponding to high salt or metal concentrations, as well as low temperature stress [4]. Anthocyanins can protect the eyes and liver, through their antioxidant and free radical elimination abilities [5].

Transcription factors such as R2R3-MYBs, bHLH, and WD40 have been shown to be involved in regulating anthocyanin biosynthesis in plants [6]. Studies have shown that bHLH transcription factors are (directly or indirectly) involved in a number of biological regulatory processes. These factors can participate in various physiological processes, such as secondary metabolism, signal transduction and growth, and development. bHLH transcription factors involved in anthocyanin biosynthesis have been reported in *Arabidopsis* [7], petunia [8,9], wheat [10], wintersweet [11], waxberry [12], gillyflower [13], and so on. The transcription factors forming the MYB–bHLH–WD40 (MBW) complex regulate anthocyanins biosynthesis [14,15]. IbbHLH2 has been identified as being involved in anthocyanin biosynthesis in purple-flesh sweet potatoes [16]. However, the involvement of transcription regulators of the promoter of *IbbHLH2* (i.e., *PIbbHLH2*) in anthocyanin biosynthesis has rarely been studied in purple-fleshed sweet potato storage roots. For this study, the transcription regulators of *PIbbHLH2* involved in anthocyanin biosynthesis were screened via yeast one-hybrid assays in purple-fleshed storage roots. In this way, IbERF1, IbERF10, and IbEBF2 are identified as the key transcription regulators of *PIbbHLH2,* regulating anthocyanin biosynthesis in purple-fleshed sweet potatoes. 

## 2. Materials and Methods

### 2.1. Plant Materials

Purple-fleshed sweet potatoes *cv.* A5 and white-fleshed sweet potatoes *cv.* Yubeibai were cultivated in the biological garden at South China Normal University, Guangzhou, GuangDong, China. Arabidopsis used for subcellular localization and dual luciferase assays were grown in a growth chamber with a day/night cycle of 16 h/8 h at 20 ± 2 °C and with a light intensity of 100 μmol/m^2^/s.

### 2.2. Extraction of Genomic DNA and RNA, Gene Isolation, and Sequence Analysis

Root tissue samples (0.5g) were ground into a fine powder in the presence of liquid nitrogen using a mortar and pestle. A plant genomic DNA kit (Cat. No. 4992201, Tiangen, Beijing, China) was used for DNA extraction and a Hipure Plant RNA Mide kit (Cat. No. R4152, Magen, Guangzhou, China) was used for RNA extraction. In order to eliminate any possibility of DNA contamination, total RNA was treated with DNAse I digestion, using an RNAse-free kit (TaKaRa, Osaka, Japan). Using a BioPhotometer plus (Eppendorf, Hamburg, Germany), DNA concentrations and purity were determined using absorbance measurements at 230, 260, and 280 nm. DNA samples were characterized by 1.8 ≤ OD_260_/OD_280_ ≤ 1.9 and OD_260_/OD_230_ ≥ 2.0, while RNA samples presented 1.9 ≤ OD_260_/OD_280_ ≤ 2.0 and OD_260_/OD_230_ ≥ 2.0.

cDNA synthesis was performed using a GoScript^TM^ Reverse Transcription System (cat. no. A5001, Promega). The promoter and genomic fragments were cloned and sequenced. PCR products were analyzed on 1.2% agarose gels. The fragment was ligated into the plasmid, transformed into *Escherichia coli* DH5α competent cells (Weidi, Shanghai, China) and sequenced at Sangon Biotech.

### 2.3. Yeast One-Hybrid Screening

The Y1H screening assays were performed using a Matchmaker Gold Yeast One-Hybrid Library Screening System. Total RNA was isolated from the storage roots of *cv*. A5 to construct the prey cDNA library (TaKaRa). A cDNA pool was separately inserted into the the prey vector pGADT7-Rec. Two amplifications of *IbbHLH2* promoter fragments (−991 bp to −531 bp and −530 bp to −1 bp) were inserted into pAbAi to construct the pAbAi-bait. The pAbAi-bait plasmids were linearized and transformed into Y1HGold. The colonies were selected on selective synthetic dextrose medium in the absence of uracil. After determining the minimal inhibitory concentrations of aureobasidin A (AbA) for the bait strains, the linear pGADT7-Rec vector was co-transformed into the bait yeast strains and selected on synthetic dextrose (SD)/-Leu/AbA plates. The primers used for the yeast one-hybrid screening are listed in Appendix A.

### 2.4. Yeast One-Hybrid Assay(y1h) Assay

Y1H assays were performed utilizing the Matchmaker Gold Yeast One-Hybrid System in order to identify that IbERF1, IbERF10, IbEBF2, IbPDC, and IbPGP19 interacted with the *IbbHLH2* promoter. The promoter of *IbbHLH2* was inserted into pAbAi to construct the pAbAi-bait. The complete CDSs of the IbERF1, IbERF10, IbEBF2, IbPDC, and IbPGP19 were separately inserted into the pGADT7 vector to construct the prey-AD vectors. The AD-prey vectors were transferred into the bait strain and grown on SD/-Leu/AbA plates. The primers used for Y1H are listed in Appendix A.

### 2.5. Yeast Two-Hybridassay (Y2H) Assay

The transcriptional activities of IbERF1, IbERF10, IbEBF2, IbPDC, and IbPGP19 were investigated using a Matchmaker^TM^ Gold Yeast Two-Hybrid System (Clontech). The full-length coding sequences of *IbERF1*, *IbERF10*, *IbEBF2*, *IbPDC,* and *IbPGP19* were cloned into the pGBKT7 vector to construct the bait-BD vectors. The PGBKT7-bait and PGADT7-empty vectors were co-transferred into the yeast strain Y2HGold and grown on the following plates: SD/-Trp medium, SD/-His-AbA medium, and SD/-His-AbA X-a-Gal plus medium. The pGBKT7-53 vector co-transformed with the pGADT7-53 vector acted as a positive control, while the pGBKT7 vector co-transformed with the pGADT7-53 vector acted as a negative control. The primers used for Y2H are listed in Appendix A.

### 2.6. Dual-Luciferase Assay 

Dual-luciferase assays were carried out to measure the transactivation of activities of IbERF1, IbERF10, IbEBF2, IbPDC, and IbPGP19 on the promoter of *IbbHLH2*. In brief, the full-length cDNAs of *IbERF1*, *IbERF10*, *IbEBF2*, *IbPDC,* and *IbPGP19* were inserted into the pGreen II 0029 62-SK vector and the promoter of *IbbHLH2* was inserted into pGreen II 0800-LUC vector. Both constructs were transformed into *Arabidopsis* protoplasts. The ratio of LUC and REN enzyme activities was measured using an E1910 Dual-Luciferase^®^ Reporter Assay System (Promega). For each interaction between transcription factors and promoters, three independent experiments were carried out, with three replicates in each experiment. A luciferase gene from Renilla driven by a 35S promoter in the luciferase vector acted as a positive control. Mixtures that contained each transcription factor and the empty vector 62-SK were also tested on the promoter as a control. The primers used for the dual-luciferase assay are listed in Appendix A.

### 2.7. Sub-Cellular Localization Analysis 

For the sub-cellular localization analysis, the upstream transcription factor CDSs without the stop codon were amplified and cloned into the pCambia1300 vector with *BamH* Ⅰ and *Hind* Ⅱ, which contained the UBQ promoter and GFP gene. Both constructs were transformed into *Arabidopsis* protoplasts. GFP fluorescence was observed using a Zeiss confocal microscope LSM710. The primers used for sub-cellular localization analysis are listed in Appendix A.

### 2.8. Real-Time Quantitative PCR

The expression of transcription regulators (*IbERF1*, *IbERF10*), transcription factors (*IbMYB1*, *IbbHLH2*, *IbWD40*), and structure genes (*IbCHI*, *IbCHS*, *IbF3H*, *IbF3′H*, *IbDFR*, *IbANS*, *IbUF3GT*) in fibrous roots, thick roots, and storage roots of purple-fleshed sweet potatoes *cv.* A5 and white-fleshed sweet potatoes *cv.* Yubeibai were analyzed using real-time quantitative PCR. First-strand cDNA was synthesized from total RNA using Prime Script™ RT Master Mix (Takara, Osaka, Japan). RT-qPCR was conducted using SYBR^®^ Premix Ex Taq™ II (Takara, Osaka, Japan) in a total reaction volume of 20 μL, consisting of 100 ng of template cDNA, each primer at 0.5 μM, and 10 μL of SYBR^®^ Premix Ex Taq™ II. The amplification program was as follows: 1 cycle of 95 °C for 10 s, followed by 40 cycles of 95 °C for 5 s and 60 °C for 30 s in a Bio-Rad CFX96 Real-Time PCR system (BIO-RAD, Hercules, California, USA), according to the manufacturer’s instructions. IbG14 was used as an internal control and calculated using the comparative Ct analysis method. The primers used for RT-qPCR are listed in Appendix A.

### 2.9. Statistical Analysis 

Three biological replicates of each sample were subjected to one-way analysis of variance (ANOVA). Significant differences were calculated using the SPSS 21.0 Statistical software (SPSS Inc., Chicago, IL, USA) using Tukey’s honest significance test (*p* < 0.05). 

## 3. Results

### 3.1. Screening the Transcription Regulators on the Promoter of IbbHLH2

Two amplifications of *IbbHLH2-1/2* promoter fragments were ligated into the pAbAi vector to generate the pAbAi-bait plasmid. The minimal inhibitory concentrations of Aureobasidin A (AbA) for the bait strains were tested according to the system user manual. The results revealed that the minimal inhibitory concentration of AbA for pAbAi–*PIbbHLH2-1* strains was 700 ng/mL, while the minimal inhibitory concentration of AbA for the positive strains (pGADT7-53+p53-AbAi) was 900 ng/mL. The self-activation of *PIbbHLH2-2* could not be suppressed due to the minimal inhibitory concentration of AbA for the pAbAi–*PIbbHLH2-2* strains being greater than 1000 ng/mL.

A total of 474 positive colonies of *PIbbHLH2-1* were screened, while 222 binding proteins of *PIbbHLH2-1* were screened in the yeast one-hybrid assay. The results for the analysis of the gene sequences of *PIbbHLH2-1* are provided in Appendix A. The proteins IbERF1, IbERF10, IbEBF2, IbPDC, and IbPGP19 interacted with the promoter of *IbbHLH2-1,* which is involved in anthocyanin biosynthesis. 

### 3.2. Transcriptional Activity of IbERF1, IbERF10, IbEBF2, IbPDC, and IbPGP19

Y2H assays were performed to identify the transcriptional activities of IbERF1, IbERF10, IbEBF2, IbPDC, and IbPGP19. The results revealed that pGADT7-53+pGBKT7-53 and pGBKT7-transcription regulators+pGADT7-empty transformed strains could grow on SD/-Trp, SD/-His-AbA plus, and SD/-His-AbA X-a-Gal plus plates, with the color of the yeast colony observed to be blue (Figure 1). Meanwhile, the pGADT7-53+PGBKT7-transformed strains did not grow on SD/-Trp, SD/-His-AbA plus, or SD/-His-AbA X-a-Gal plus plates. These results indicate that IbERF1, IbEBF2, and IbERF10 were responsible for the transcriptional activity.

### 3.3. Interaction between Transcription Regulators and PIbbHLH2-1

Y1H assays were performed to identify whether the transcription regulators (i.e., IbERF1, IbERF10, IbEBF2, IbPDC, IbPGP19, IbUR5GT, and IbDRM) interacted with *PIbbHLH2*-1. The Y1H results indicated that the positive control grew on SD/-Leu/AbA plates, while the negative control could not grow on SD/-Leu/AbA plates. All of the transformed strains grew on SD/-Leu/AbA plates (Figure 2). The results served to identify the interaction between transcription regulators and *PIbbHLH2-1*. 

Dual-luciferase assays were carried out to confirm the interactions between the transcription regulators and *PIbbHLH2-1*. As illustrated in Figure 3, the transcription regulators presented a significant activation effect on the *PIbbHLH2-1*. The results indicated interactions between the transcription regulators (i.e., IbERF1, IbERF10, IbEBF2, IbPDC, IbPGP19, IbUR5GT, and IbDRM) and *PIbbHLH2-1*. 

### 3.4. Sub-Cellular Localization of the Transcription Regulators

Green fluorescent protein (GFP) was fused to the C-terminus of transcription regulators and transiently expressed in *Arabidopsis* protoplasts under the control of the UBQ promoter for sub-cellular localization assays. The green fluorescence in the GFP control was observed in the nucleus and cytoplasm. Green fluorescence was only detected in the nucleus in *IbERF1/IbERF10/IbPDC/IbPGP19*-GFP fusions (Figure 4), suggesting that IbERF1, IbERF10, IbPDC, and IbPGP19 are nuclear proteins. Meanwhile, the proteins IbUR5GT, IbDRM, and IbEBF2 were expressed in both the nucleus and cytoplasm.

### 3.5. Expression Characteristics of Transcription Regulators

The gene expression of transcription regulators, transcription factors, and structural genes involved in anthocyanin biosynthesis in different root stages of purple- and white-fleshed sweet potatoes was analyzed via real-time PCR (Figure 5). The results indicated that the expression levels of transcription factors and structural genes in different root stages of purple-fleshed sweet potatoes were higher than those in white-fleshed sweet potatoes. The expression of *IbERF1* in different root stages of purple-fleshed sweet potatoes were lower than that in white-fleshed sweet potatoes, indicating that *IbERF1* negatively regulates anthocyanin accumulation. The expression of *IbERF10* in different root stages of purple-fleshed and white-fleshed sweet potatoes did not present a positive or negative relationship with the anthocyanin accumulation, indicating a more complex mechanism in the regulation of anthocyanin biosynthesis.

## 4. Discussions

Anthocyanin biosynthesis is regulated by the MYB–bHLH–WD40 complex in plants. Other TFs, including ERFs, WRKYs, COP1, and NACs proteins, have been identified to play a role in the regulation of anthocyanin biosynthesis [17,18,19]. In this study, yeast one-hybrid screening was carried out to screen the transcription regulators of *PIbbHLH2,* which are involved in anthocyanin biosynthesis. The results revealed IbERF1 and IbERF10 as transcription factors acting on *PIbbHLH2-1* to regulate anthocyanin biosynthesis. 

The ethylene response factors AP2/ERF are some of the largest transcription factors, which are involved in multiple physiological processes, including plant growth and responses to biological and non-biological stress [20,21]. Studies have shown that ERF transcription factors regulate anthocyanin biosynthesis in plants; for example, in *Salvia miltiorrhiza* flowers, the transcription factors AP2/ERFs may regulate anthocyanin biosynthesis, according to the significantly different expression levels between purple and white flowers observed in transcriptome data [22]. ERF transcription factors have been identified to regulate ethylene- and light-induced anthocyanin biosynthesis in different species. In plum, seven PsERFs have been positively correlated with PsMYB10 and most of the structural genes involved in anthocyanin biosynthesis [23]. In *Arabidopsis*, a double mutant (*aterf4* and *aterf8*) presented reduced light-induced anthocyanin accumulation, indicating that AtERF4 and AtERF8 positively regulate anthocyanin biosynthesis [24]. In pears, ERF transcription factors have been shown to be involved in light-induced anthocyanin biosynthesis [25].

ERFs interact with MYB and bHLH proteins and also bind to the promoters of MYB to regulate anthocyanin biosynthesis in plants. In apples, MdERF1B has been observed to interact with MdMYB9, MdMYB1, and MdMYB11 proteins to regulate anthocyanin biosynthesis. Furthermore, MdERF1B also bind to the promoters of *MdMYB9*, *MdMYB1*, and *MdMYB11* [26]. MdERF38 have been shown to bind to the promoters of *MdMYB1* to regulate drought stress-induced anthocyanin biosynthesis at the post-translational level [27]. In pears, Pp4ERF24 and Pp12ERF96 regulated light-induced anthocyanin biosynthesis [28]. In strawberries, FaERF9 and FaMYB98 formed and ERF–MYB complex and activated the FaQR promoter, increasing furaneol content in cultivated strawberries [29]. In this study, the proteins binding *PIbbHLH2-1* were screened through yeast one-hybrid assays. The proteins IbERF1, IbERF10, IbEBF2, IbPDC and IbPGP19 interacted with the promoter of *IbbHLH2-1,* which is involved in anthocyanin biosynthesis. ERF proteins encode a member of the ethylene response factor family, which contains one AP2 domain, as well as an F-box protein involved in the ubiquitin/proteasome-dependent proteolysis of EIN3. The PDC protein encodes a member of the pyruvate decarboxylase family of proteins, while the PGP protein encodes an ATP-binding cassette transporter. In addition to the yeast one-hybrid assays, the interactions between the transcription factors and PIbbHLH2-1 were verified via a dual-luciferase assay. The nuclear localization of IbERF1 and IbERF10 was confirmed by the transient expression of GFP fused to the C-terminus of TFs in *Arabidopsis* protoplasts. Overall, IbERF1 and IbERF10 interacted with the promotor of *IbbHLH2-1* to regulate anthocyanin biosynthesis in purple-fleshed sweet potatoes. The results of this study provide a possible underlying mechanism of anthocyanin biosynthesis, potentially allowing for a better understanding of the regulatory network in purple-fleshed sweet potatoes.

## 5. Conclusions

The upstream transcription factors of PIbbHLH2 were screened via yeast one-hybrid screening in purple-fleshed sweet potato storage roots. The results indicated the proteins IbERF1, IbERF10, IbEBF2, IbPDC and IbPGP19 as upstream transcription factors of PIbbHLH2 to regulate anthocyanin biosynthesis in purple-fleshed sweet potatoes. Furthermore, the sub-cellular localization analysis indicated that IbERF1, IbERF10, IbPDC, and IbPGP19 were nuclear proteins. Finally, IbERF1 and IbERF10 were screened as the transcription factors of *PIbbHLH2-1*, regulating anthocyanin biosynthesis in purple-fleshed sweet potatoes. These results describe a possible underlying mechanism of anthocyanin biosynthesis, potentially allowing for better understanding of the regulatory network in purple-fleshed sweet potatoes. In the future, the analysis of the responses of these transcription factors to environmental factors is expected to provide a basis for understanding the mechanism of anthocyanin accumulation in underground root tubers.

## Figures and Tables

**Figure 1 genes-14-01042-f001:**
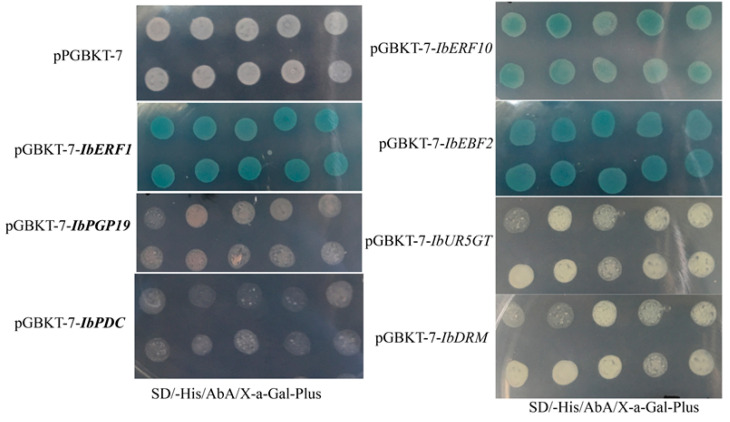
Transcriptional activation of transcription regulators in yeast cells. The Y2H Gold strains were successfully transformed with corresponding vectors and grown on SD/-His/AbA/X-α-Gal plates at 30 ℃ for 3–5 days. Ten points for each treatment were grown on each Petri dish. Transcriptional activation was monitored, according to the growth status of yeast cells, using the X-α-Gal assay.

**Figure 2 genes-14-01042-f002:**
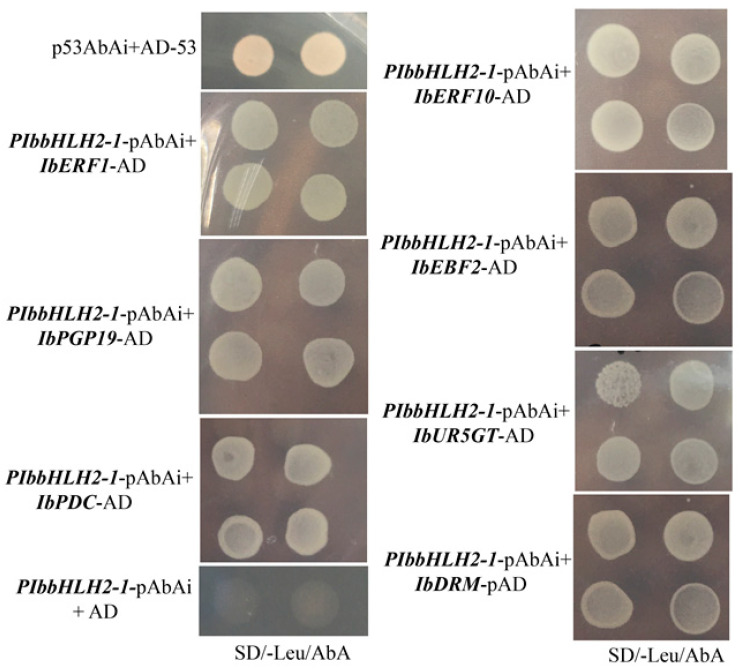
Determination of the interactions between transcription regulators and the promoter of *IbbHLH2-1* through yeast one-hybrid assays. Y1H Gold strains successfully transformed with corresponding vectors were grown on the.SD/-Leu/AbA plates at 30 ℃ for 3–5 days. Four points of each treatment were grown on each Petri dish. Interactions were confirmed according to the growth status of yeast cells. AD: pGADT-7.

**Figure 3 genes-14-01042-f003:**
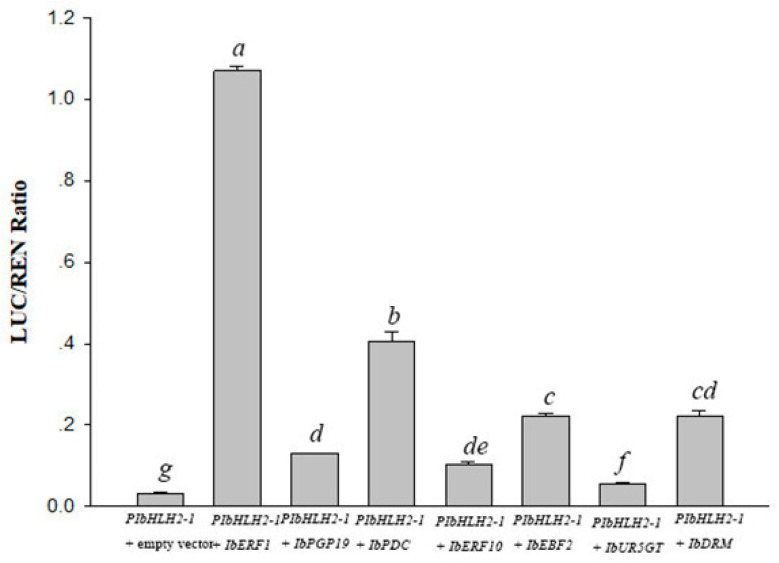
The IbERF1, IbERF10, IbEBF2, IbPDC, and IbPGP19 proteins activated the promoter of IbbHLH2-1 in dual-luciferase assays. Error bars represent the standard deviation (SD). The significance tests are denoted by a–g. Different lowercase letters in columns indicate significant differences (*p* < 0.01).

**Figure 4 genes-14-01042-f004:**
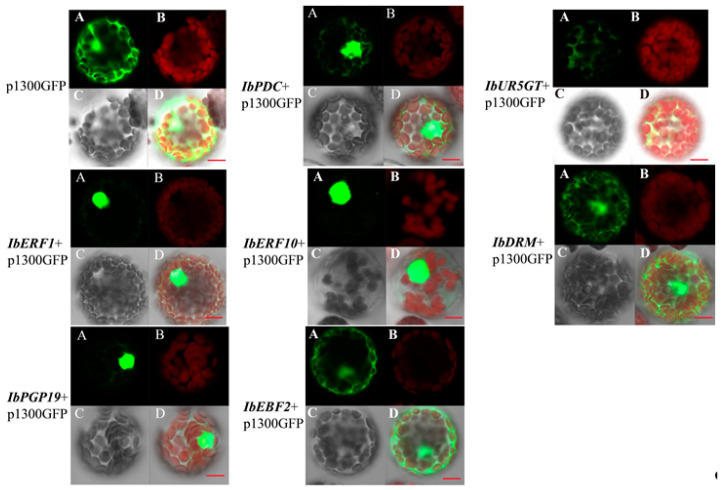
Sub-cellular localization of transcription regulators in *Arabidopsis* protoplasts. The fusion protein and GFP control were transiently expressed in *Arabidopsis* protoplasts. Bars represent 20 µm. A, GFP; B, Chloroplast; C, Light field; and D, Merged graph.

**Figure 5 genes-14-01042-f005:**
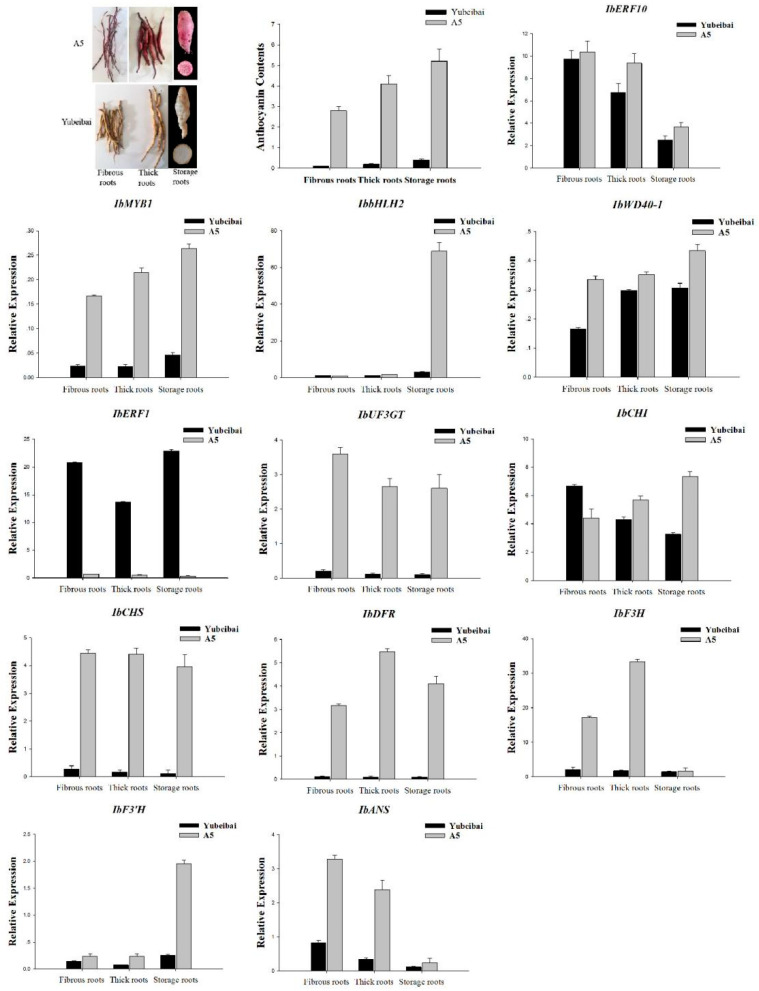
Relative expression levels of transcription regulators, transcription factors, and structural genes involved in anthocyanin biosynthesis at different root stages in purple- and white-fleshed sweet potatoes. Fibrous roots (diameter < 2 mm), Thick roots (2 mm < diameter > 5 mm), and Storage roots (diameter > 5 mm).

## Data Availability

Not applicable.

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
