# Peer review of "Yeast One-Hybrid Screening for Transcription Factors of IbbHLH2 in Purple-Fleshed Sweet Potato"

_genes, 2023, doi:10.3390/genes14051042_

Round 1

Reviewer 1 Report

Review 12.04

In the manuscript entitled “Yeast one-hybrid screening for transcription factors of IbbHLH2 in purple-fleshed sweet potato (Ipomoea batatas [L] 3 Lam.)” the authors revealed that  the transcription factors IbERF1 and IbERF10 belonging to AP2/ERF family were able to directly bind to transcription factor PIbbHLH2 which is involved in anthocyanin biosynthesis in purple-fleshed sweet potato. In addition to yeast one-hybrid assay, the interaction between the transcription factors and PIbbHLH2-1 was verified by dual-luciferase assay. The nuclear localization of IbERF1 and IbERF10 was confirmed by the transient expression of GFP fused to the C-terminus of Tfs in Arabidopsis protoplasts.  The authors also examined the expression patterns of several genes required for anthocyanin biosynthesis at different stages of root development in purple-fleshed and white-fleshed sweet potato.

Overall, the authors present convincing results confirming the engagement of ethylene response factors and bHLH in anthocyanin biosynthesis in Ipomoea batatas obtained on other plant species. As extensive editing of the English language and style is required to make it more comprehensible. I would also recommend redesigning figure 5, since the scale makes the captions in the figures indistinguishable even when enlarged. Anthocyanin content data is also needed. The authors should give an extended designation and a brief description of all the genes under consideration in order to make the manuscript more understandable for readers. In addition, a more detailed discussion of the results obtained is recommended. Pictures of plant phenotypes should be added as well.

Low level of English

Author Response

Dear Editors and Reviewers of Journal Genes:

Thank you very much for providing the thorough and excellent reviews on our manuscript genes-2355040 entitled "Yeast one-hybrid screening for transcription factors of IbbHLH2 in purple-fleshed sweet potato (Ipomoea batatas [L] Lam.)". The comments have been very helpful for revising and improving our paper, as well as providing guidance for our future research. We have studied the comments and the annotated manuscript carefully and made corrections which we hope meet with your approval. The point-by-point response to editor and review’s questions were presented as follows:

Reviewer: 1

  1. Comment:As extensive editing of the English language and style is required to make it more comprehensible.

Respond: Done. The manuscript language had been edited by editing service agency and the certificate were provided in the appropriate location.

  1. Comment:I would also recommend redesigning figure 5, since the scale makes the captions in the figures indistinguishable even when enlarged. Anthocyanin content data is also needed. Pictures of plant phenotypes should be added as well.

Respond: Done (Line 225-226). We had redesigned figure 5, added the anthocyanin content data and plant phenotypes pictures.

  1. Comment:The authors should give an extended designation and a brief description of all the genes under consideration in order to make the manuscript more understandable for readers.

Respond: Done (Line 264-268). We had added an extended designation and a brief description of all the genes under consideration.

  1. Comment: In addition, a more detailed discussion of the results obtained is recommended.

Respond: Done (Line 261-271). We had added more detailed discussion about the results obtained.

Special thanks to you for your good comments.

Reviewer 2 Report

The study by Fu et al., aimed to identify the transcription regulators of IbbHLH2, a transcription factor known to be involved in anthocyanin biosynthesis in purple-flesh sweet potato. The researchers identified proteins that bind to the promoter of IbbHLH2 and confirmed the interaction between these proteins and the promoter using luciferase reporter assays and yeast one-hybrid. While the study identifies multiple regulators of IbbHLH2, it does not emphasize the relevance of the study with respect to sweet potato research and how it impacts future research in the crop.

Major concerns:
While the discussion provides a comprehensive review of the literature on transcriptional regulation of anthocyanin biosynthesis, it lacks emphasis on the specific findings of the current study. Only a small portion of the discussion directly relates to the authors' research, with the majority of the discussion focusing on previous research in the field. As a result, the authors could benefit from providing a more detailed and specific interpretation of their findings and discussing the significance of their research in the broader context of the field.

Some of the figure legends are very short and not stand alone. They can be improved to explain the figure in greater detail. For instance, in Figure 3 legend, “Transcription regulators activates the promoters of IbbHLH2-1 in dual-luciferase assays. Error bars represent standard deviation (SD). The significance tests are shown as a, b, c, d. Completely different lowercase roman alphabet on the column chart indicates significant differences (P<0.01)”, what do the authors mean by “completely different lowercase roman alphabet”? “The significance tests are shown as a, b, c, d.” There are other alphabets as well. It is very confusing. It is good to have a statement to each figure legend summarising the result of the figure, and then go onto explain the figure adequately.

The manuscript requires extensive language editing. I have mentioned some of the errors in the minor concerns. But there are grammatical and typographical errors throughout the manuscript.

Minor concerns:

Line 11: “involed” to “involved”

Line 11: What does the “P” in “PIbbHLH2” represent? Please add the explanation in the abstract as well as at the first occurrence in the main text as well.

Line 12: What do the authors mean by “...had rarely been studied”?

Line 40: “biosynthesisin”?

Line 45: “involed”?

Line 73: “screenin”?

Line 123: “The expression of transcription regulators, transcription factors (IbMYB1, IbbHLH2, IbWD40)...” What are the transcriptional regulators?

Line 159: “The pGADT7-53+PGBKT7 transformed strains can not grown on SD/-Trp, SD/-His-AbA plusand SD/-His-AbA X-a-Gal plus plate.” Please correct spelling and grammatical errors.

Lines 168-173: Correct all language errors.

Figure 5: The resolution of this figure is very poor. This figure needs to be replaced.

Is the “5. Conclusions” section required? If yes, this section is poorly written and must be improved significantly.

The manuscript requires extensive language editing. I have mentioned some of the errors in the minor concerns. But there are grammatical and typographical errors throughout the manuscript.

Author Response

Reviewer: 2

1.Comment: While the discussion provides a comprehensive review of the literature on transcriptional regulation of anthocyanin biosynthesis, it lacks emphasis on the specific findings of the current study. As a result, the authors could benefit from providing a more detailed and specific interpretation of their findings and discussing the significance of their research in the broader context of the field.

Respond: Done (Line 261-275). We had added more detailed and specific interpretation of our findings and discussing the significance of our research in the broader context of the field.

2.Comment: Some of the figure legends are very short and not stand alone. They can be improved to explain the figure in greater detail. For instance, in Figure 3 legend, “Transcription regulators activates the promoters of IbbHLH2-1 in dual-luciferase assays. Error bars represent standard deviation (SD). The significance tests are shown as a, b, c, d. Completely different lowercase roman alphabet on the column chart indicates significant differences (P<0.01)”, what do the authors mean by “completely different lowercase roman alphabet”? “The significance tests are shown as a, b, c, d.” There are other alphabets as well. It is very confusing. It is good to have a statement to each figure legend summarising the result of the figure, and then go onto explain the figure adequately.

Respond: Done (Line 173-177, Line 186-190, Line 197-200, Line 211-213, Line 227-230). We had modified statement of figure legends according to the suggestion.

  1. Comment:The manuscript requires extensive language editing. I have mentioned some of the errors in the minor concerns. But there are grammatical and typographical errors throughout the manuscript.

Respond: Done. The manuscript language had been edited by editing service agency and the certificate were provided in the appropriate location.

4.Comment: Line 11: “involed” to “involved”

Respond: Done (Line 12, Line 47, Line 49, Line 228). We had modified all “involed” to “involved” in manuscript.

5.Comment: Line 11: What does the “P” in “PIbbHLH2” represent? Please add the explanation in the abstract as well as at the first occurrence in the main text as well.

Respond: Done (Line 11, Line 13, Line 15, Line 20-21, Line46). We had added the explanation in the abstract and at the first occurrence in the main text.

6.Comment: Line 12: What do the authors mean by “...had rarely been studied”?

Respond: Done (Line 11-12). We had modified the sentence as follows: However, little is known about the upstream transcription regulators on the promoter of IbbHLH2 involved in anthocyanin biosynthesis.

7.Comment: Line 40: “biosynthesisin”? Line 73: “screenin”?

Respond: Done (Line 42, Line 76). We had modified “biosynthesisin” to “biosynthesis” and “screenin” to “screening”.

8.Comment: Line 123: “The expression of transcription regulators, transcription factors (IbMYB1, IbbHLH2, IbWD40)...” What are the transcriptional regulators?

Respond: Done (Line 127). We had added the transcriptional regulators as follows: The expression of transcription regulators (IbERF1, IbERF10).

9.Comment: Line 159: “The pGADT7-53+PGBKT7 transformed strains can not grown on SD/-Trp, SD/-His-AbA plusand SD/-His-AbA X-a-Gal plus plate.” Please correct spelling and grammatical errors.

Respond: Done (Line 205). We had modified the sentence as follows: Meanwhile, the pGADT7-53+PGBKT7-transformed strains did not grow on SD/-Trp, SD/-His-AbA plus, or SD/-His-AbA X-a-Gal plus plates. 10.Comment: Lines 168-173: Correct all language errors.

Respond: Done. The manuscript language had been edited by editing service agency and the certificate were provided in the appropriate location.

11.Comment: Figure 5: The resolution of this figure is very poor. This figure needs to be replaced.

Respond: Done. We had redesigned figure 5, added the anthocyanin content data and plant phenotypes pictures.

12.Comment: Is the “5. Conclusions” section required? If yes, this section is poorly written and must be improved significantly.

Respond: Done (Line 270-281). We had improved the section in conclusions.

Special thanks to you for your good comments.

Thank the editors and reviewers again for their time and comments and believe the manuscript has been greatly improved as a consequence.

Yours sincerely,

Feng Gao

Round 2

Reviewer 1 Report

 Review  v2 genes 2355040

After intensive editing of the language and additions made, the manuscript “of Fu and colleagues “Yeast one-hybrid screening for transcription factors of 2 IbbHLH2 in purple-fleshed sweet potato” basically complies with accepted standards and can be accepted for publication. However, some corrections still need to be made, especially in the added passages.

Minor concerns:

Line 34: «heavil” to “heavily”

Line 47:” regulates» to “regulate

Line 101: “Yeast two-hybridassay (Y2H) assay”. to “Yeast two-hybrid (Y2H) assay“

Line 168:” activity» to “activities”

Line 227: “were” to “was”

Line 234: It is necessary to remove Figure 5 from the previous version of the manuscript

Lines 247-249: The ethylene response factors AP2/ERF are sonme of the largest transcription factors, which are involved in multiple physiological processes including plant growth and 248 responses to biological and non-biological stress2 to

“The ethylene response factors AP2/ERF are involved in multiple physiological processes including plant growth and responses to biological and non-biological stress”

Or

“The ethylene response factors AP2/ERF belong to the largest family of TFs involved in multiple physiological processes including plant growth and responses to biological and non-biological stress”

Lines 247-249:  at the post-translational level [27}”  to “at both transcriptional and post-translational levels[27}”

Line 274:” ERF proteins encode a member of the ethylene response factor family which contains one AP2 domain”, to “ERF proteins encode members of the ethylene response factor family which contain one AP2 domain”

Line 294: “to regulating” to  “ regulating”  or  “to regulate”

Minor corrections  and text editing are needed